materials science/solid-state physics/chemical physics

inkjet printing, suspension inks, solution inks, oxide films, morphology

**Authors for correspondence:**
Mei Fang
e-mail: meifang@csu.edu.cn
Lyubov Belova
e-mail: lyuba@kth.se

# Design and tailoring of inks for inkjet patterning of metal oxides

Mei Fang[1,2], Tianli Li[1], Sangjian Zhang[1], K. V. Rao[2] and Lyubov Belova[2]

[1]Hunan Key Laboratory of Super Microstructure and Ultrafast Process, School of Physics and Electronics, Central South University, Changsha, Hunan 410083, People's Republic of China
[2]Department of Materials Science and Engineering, KTH-Royal Institute of Technology, Stockholm SE10044, Sweden

 MF, 0000-0002-0995-1190; LB, 0000-0003-4889-4210

Inkjet printing has become a promising, efficient, inexpensive, scalable technique for materials deposition, mask-less and digital patterning in many device applications. Meanwhile, the ink preparation remains a challenge especially for printing functional oxide materials. Based on the principles of inkjet printing (especially relevant for piezoelectric drop-on-demand inkjet printer) and the process of the conversion of liquid ink into solid thin films of oxide materials, we present two approaches to the design and tailoring of inks: (i) oxide particle suspensions (e.g. $SiO_2$, $TiO_2$, $Fe_3O_4$) and (ii) metal-acetates precursor solutions for directly printing oxide thin films (e.g. ZnO, MgO, ITO and so forth). The solution inks are stable and produce tunable oxide films with high density and smooth surface. For some of the inks containing multi-type acetates with possible phase separation even before calcinations, we have developed a chelating procedure in order to tailor the films into single-phase homogeneity. The work lays a foundation for inkjet printing of oxides films for functional applications in electronic, photonic and energy devices.

## 1. Introduction

Nowadays, the inkjet printer is a familiar tool which is used in almost every office and household at an affordable price. In recent years, more and more efforts have been made to extend the inkjet printer into a versatile tool in industrial manufacturing processes. New markets based on inkjet technology, such as micro-electronics, medicine and pharmaceuticals, and renewable energy, have been exploited using precursor inks for 'smart' materials deposition instead of pigment inks used for graphic art printing [1,2]. It is 'smart' because the printing process and the deposition pattern can be

This article has been edited by the Royal Society of Chemistry, including the commissioning, peer review process and editorial aspects up to the point of acceptance.

controlled by a digital computer with high repeatability. The direct-patterning ability reduces waste during fabrication, and the process is cost-effective with efficient usage of precursor materials, which is friendly to environment. In addition, the printer can generate tiny volume (in picolitre range) ink droplets for deposition, ensuring the accurate deposition of minute quantity of materials as well as the resolution in sub-micrometre range for patterning. Multi-pass printing endows the inkjet technique the ability for three-dimensional prototyping [3]. Inkjet printing produces high-quality films which are comparable with physical deposition techniques involving vacuum systems and gases, such as sputtering, thermal evaporation and pulsed laser beam deposition. What is more, it can produce metre-scale homogeneous films with productivity orders of magnitude higher. It is an inexpensive waste-free fabrication tool that is easy to use and that does not require clean rooms, sophisticated large-scale facilities and extensive infrastructure. These advantages have attracted scientists' and manufacturers' attention to develop inkjet technique for fabrications of a variety of materials, like polymers, proteins, nanomaterials and quantum dots, two-dimensional materials, among others [1,4–6]. With the digitally designed patterns, inkjet technique is attractive for various applications such as electric devices [7–9], thin film transistor [4], solar cells [10–13], sensors [14,15], optical devices [2,16], displayers [17], prototyping [18] and biomedicine applications [15]. The key challenge of developing inkjet materials deposition is the ink preparation, which is very specific depending on materials and applications.

It was in 1867 that Lord Kelvin first observed the break-up of liquid stream into individual droplets, which is the basic principle for inkjet technologies developed ever since. The channel walls can be piezo-actuated and jet out the ink droplet with controllable size and shape, known as piezoelectric drop on demand (DOD) inkjet printer [19]. The deposition of the ink droplet on a substrate starts from the impact of the liquid and the solid surface. The impact dynamics depend on the surface properties of the solid surface, the impact velocity, the properties like viscosity, surface tension of the liquid ink, etc. Certain post-process is required to convert the liquid droplet into solid materials after printing, such as evaporation of the liquid, calcination of the precursors or UV-cure, depending on the precursor inks and target materials [5,20,21].

The performance of each process in inkjet printing is closely related to the properties of the ink. Physicochemical properties of the ink like stability, surface tension, viscosity and chemical compatibility with the printer system, among others, are important to be well controlled for successful printing [22]. (i) The properties of the ink should remain constant during printing for repeatable depositions. Factors like the aggregation of small size particles, sedimentation of the dispersed phases, polymerization in UV ink, etc., should be avoided in preparing a printable ink. (ii) The specified viscosity of the ink used in a DOD printer is typically in the range of 1–25 mPa s, which affects the formation and the motion of the ink droplet during printing [23,24]. (iii) The specified surface tension range of DOD inkjet ink is typically 20–50 mN m$^{-1}$. The surface tension of the ink influences the interaction between the ink droplet and the substrate and thence the final geometry of the patterns [25,26]. (iv) The ink should be chemically compatible with the printer system. Chemical reactions between the ink and the printing system can destroy the printhead and thus the printing process becomes no longer reliable. Preparation of suitable ink for printing thence continues to be a great challenge to meet these multiple requirements. Generally, inks for printing oxide materials are prepared from either particle suspensions or precursor solutions. For a suspension-based ink, the size of the oxide particles should be substantially smaller than the nozzle size to avoid nozzle clogging. The size effects on the random Brownian motions of particles in liquid as well as on the coffee-ring of printed films should be considered. Surface modification is another choice for preparing physicochemically stable inks with low sedimentation rate. For a solution-based ink, the precursor materials should be easily convertible into target film of high purity. Here, the solution-based inks are prepared from metallic acetates, since (i) they are soluble in water and many organic solvents; (ii) they can interact with organic solvents and form gel after printing to achieve uniform material deposition without coffee-ring effect [27,28]; (iii) the introduced impurity elements such as carbon and hydrogen can be easily burned off, and high-purity metal oxides can be obtained; (iv) they can convert into oxides at a low decomposition temperature; and (v) they are widely available and low-cost.

In this paper, we present our extensive research related to development of inks for printing advanced functional oxides. Based on the versatile process of drop on demand inkjet printing, we have designed and tailored inks for multiple purpose oxides printing such as TiO$_2$, ZnO, MgO and ITO, either oxide particle suspensions or precursor solutions. We first monitor different types of inks using a camera to quickly screen out the fast precipitation ones. Typically, the ink is printable if there is no visible sedimentation for at least two hours. Then, ultimately the quality of the inks is judged by the morphologies and features of the printed oxide films. Methods of improving inks for high-quality film printing are further suggested

**Table 1.** Details of different inks.

| ink | molar concentration (mol l$^{-1}$) | target film | precursor materials | solvent(s) | viscosity (mPa s) | surface tension (mN m$^{-1}$) |
|---|---|---|---|---|---|---|
| A | 0.1 | SiO$_2$ | SiO$_2$ particles | FAM | ∼3.6 | ∼58 |
| B | 0.25 | TiO$_2$ | TiO$_2$ particles | IPE | ∼2.4 | ∼28 |
| C | 0.02/0.001 | Fe$_3$O$_4$ | Fe$_3$O$_4$ nanoparticles | DI water | ∼1 | ∼73 |
| D | 0.02/0.001 | Fe$_3$O$_4$ | Fe$_3$O$_4$ nanoparticles | TMAH & DI water | ∼1 | ∼73 |
| E | 0.25 | ZnO | Zn-acetate | IPE | ∼2.4 | ∼28 |
| F | 0.25 | 10 at% Fe-ZnO | Zn-acetate & Fe(II)-acetate | IPE | ∼2.4 | ∼28 |
| G | 0.25 | 5 at% Mg-ZnO | Zn-acetate & Mg-acetate | IPE | ∼2.4 | ∼28 |
| H | 0.25 | MgO | Mg-acetate | MOE | ∼1.7 | ∼33 |
| I | 0.25 | 2 at% Fe-MgO | Mg-acetate & Fe(II)-acetate | MOE | ∼1.7 | ∼33 |
| J | 0.1 | 10 at% Sn-In$_2$O$_3$ | In-acetate & Sn-acetate | ATA & EA | ∼1 | ∼31 |
| K | 0.1 | 10 at% Sn-In$_2$O$_3$ | In-acetate & Sn-acetate | ATA, EA & H$_2$O$_2$ | ∼1 | ∼31 |

with respect to different types of inks. Specifically, we develop a chelating process to suppress possible phase separation in printed films (e.g. ITO films) when several different acetates are used in the solution ink. The work provides guidance for developing relevant inks for printing different technologically important oxide materials using a piezoelectric DOD inkjet printer.

# 2. Material and methods

## 2.1. Ink preparation

We prepare SiO$_2$, TiO$_2$, Ag-TiO$_2$ and Fe$_3$O$_4$ suspension inks using the respective nanoparticles, and acetate solutions as precursors for DOD inkjet printing ZnO, MgO, Fe-doped ZnO/MgO and ITO films. The details are presented in table 1.

Oxide particle suspensions (inks A–D) are prepared by ultrasonicating the oxide nanoparticles in a solvent. (A) SiO$_2$ spheres with different sizes are synthesized from the hydrolysis of tetraethyl orthosilicate (TEOS) [29] and dispersed in formamide (FAM). (B) TiO$_2$ particles are dispersed in 2-isopropoxyethanol (IPE) for printing. The sizes of the TiO$_2$ particles are in the range of 70–400 nm with an average value of approximately 170 nm [10,30], which are more than three orders of magnitude smaller than the nozzle size of the printhead. (C) Fe$_3$O$_4$ nanoparticles with high-saturation magnetization are prepared by co-precipitation of Fe$^{2+}$ and Fe$^{3+}$ solvent (pH value of 2) in ammonia hydroxide solution (pH value of 10.5) using a 'rapid mixing' technique [31,32]. The particles are washed and then dispersed in deionized water (DI water) for suspensions. For aqueous ferrofluids (ink D), one drop of tetramethyl-ammonium hydroxide (TMAH, 25 wt% aqueous solution) is added into the washed particle slurry and fully mixed by moving the particles around using a magnet. This aqueous ferrofluids can be dispersed in water without sediments for over a year, but retained the high-saturation magnetization [33].

Metal acetate solutions (inks E–J): (ink E) The ink precursor for *in-situ* inkjet printing ZnO films is prepared by dissolving zinc acetates in IPE. Doping of secondary elements can be achieved by adding corresponding acetates into the solvents, for example, iron acetate and magnesium acetate for the inks

of printing Fe-doped ZnO (ink F) and Mg-doped ZnO (ink G), respectively. Similarly, the inks for inkjet printing of MgO (ink H) and Fe-doped MgO (ink I) are prepared by dissolving magnesium acetate and iron acetate in 2-methoxyethanol (MOE). (ink J) Indium acetate and tin acetate are dissolved in acetylacetone (ATA, viscosity $\eta$: approx. 0.8 mPa s, surface tension $\gamma$: approx. 31 mN m$^{-1}$) for inkjet printing indium tin oxide (ITO). The viscosity and the surface tension of the final inks can be tuned by adding a few drops of ethanolamine (EA, $\eta$: approx. 24.1 mPa s, $\gamma$: approx. 48 mN m$^{-1}$) for successful printing.

Acetates chelated with the solvent (ink K): The ink for printing uniform ITO films is prepared by first dissolving the acetates in ATA solvent and then heating the solution at 120°C, and 0.01 mol hydrogen peroxide (30 wt% in water) is added dropwise in six aliquots at 30 min intervals into the heated solution. The acetates are chelated with the solvent and thereby suppress phase separation during drying and calcinating [7].

## 2.2. Film printing and characterizations

The films are printed on well-cleaned silicon and glass substrates (first by ultrasonicating in acetone and then in ethanol, followed by rinsing in DI water and blow dried by nitrogen gun) preheated at 60°C–80°C. The as-printed films are dried on a 180°C hotplate and then sintered/calcinated in an oven at certain temperature (300°C–700°C) for target oxide films. The obtained films are further characterized by optical microscope, high-resolution scanning electron microscope (SEM, Nova 600 Nanolab) and atomic force microscope (AFM, JSPM-5400) for investigations of morphologies and features of the films. X-ray diffractometer (XRD, Siemens D5000) is used to study the structure, and the magnetic properties and electric properties of the films are characterized by a superconducting quantum interference device (SQUID, Quantum Design) and a home-designed set-up using the standard four-probe method, respectively.

# 3. Results and discussion

## 3.1. Ageing of inks

Some inks can exhibit ageing over time. We visually monitor the ink at different time intervals with a digital camera to quickly screen out the fast sedimentation inks. Figure 1 presents visual images of the fresh and aged inks at intervals of time scales beyond a month before inkjet printing the respective oxide films. The results show that:

(i) The oxide particles in suspension inks (inks A–C, figure 1a–d) settle sooner or later, and the sedimentation rate depends on the size of the particles. This is because the random Brownian motion of smaller particles in liquid is more strenuous [34,35]. The sediments can be re-dispersed into the solvent by ultrasonication.

(ii) For $Fe_3O_4$ particles, even though they are nanosized [36], magnetic interactions among the particles accelerate the aggregation process. The ink shows sediments in a few hours of ageing (ink C, figure 1d). By adding one drop of TMAH, the hydroxide anions attach to the surface of $Fe_3O_4$ nanoparticles and the tetramethyl-ammonium cations form a shell for the nanoparticles. The electrostatic repulsion between the cations prevents the nanoparticles from aggregation and sediment in the ink [33], and stable inks can be obtained known as aqueous ferrofluid (ink D, figure 1e) which maintain the dispersion for years without sediment [31]. These particle suspension inks can be used for printing porous films, for example, $TiO_2$ films for dye-sensitized solar cell applications [10,37], or artificial photonic crystals for optical applications (e.g. $SiO_2$ photonic crystals [29]).

(iii) The acetate solutions (inks E–J, figure 1f–k) are uniform and transparent without sediment over years.

(iv) Comparing with the yellowish acetate solution (ink J, figure 1k) for printing ITO thin films, the corresponding chelated ink (ink K, figure 1l) is red-brown because of the formation of acetylacetonate complexes and shows sediments during ageing. The aged ink can be re-treated in a hot-water (70–90°C) bath to obtain the properties of fresh ink. These acetates-based solution inks can produce high-quality films with tunable chemical composition and microstructures, which can be used for printing electronic devices [7,33].

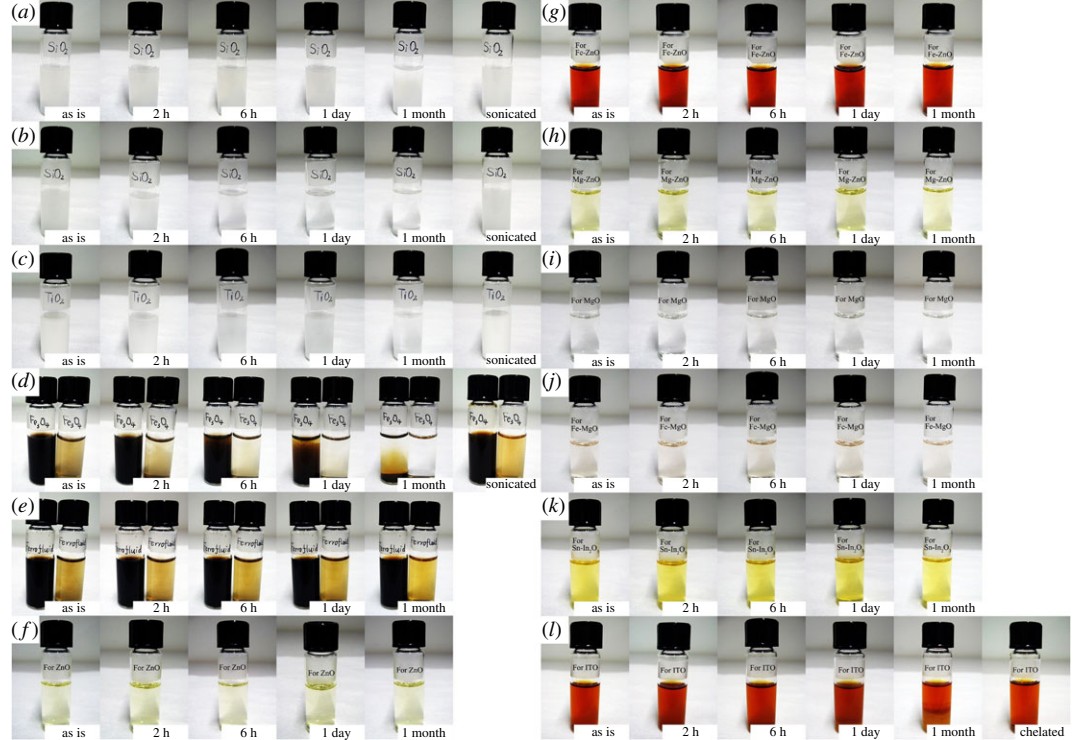

**Figure 1.** Ageing of inks prepared from different routes for printing oxide films. (*a*) Ink A1: 0.1 mol l$^{-1}$ SiO$_2$ in FAM solvent, with particle size of approximately 260 nm. (*b*) Ink A2: 0.1 mol l$^{-1}$ SiO$_2$ in FAM solvent, with particle size of approximately 520 nm. (*c*) Ink B: 0.25 mol l$^{-1}$ TiO$_2$ in IPE solvent. The particle size is in the range of 70–400 nm. (*d*) Ink C: 0.02 and 0.001 mol l$^{-1}$ Fe$_3$O$_4$ nanoparticle suspensions in DI water. (*e*) Ink D: 0.02 and 0.001 mol l$^{-1}$ Fe$_3$O$_4$ aqueous ferrofluids. (*f*) Ink E: 0.25 mol l$^{-1}$ [Zn] acetate in IPE solvent. (*g*) Ink F: 0.25 mol l$^{-1}$ acetates in IPE solvent with 10 at% [Fe] in [Zn] + [Fe]. (*h*) Ink G: 0.25 mol l$^{-1}$ acetates in IPE solvent with 5 at% [Mg] in [Zn] + [Mg]. (*i*) Ink H: 0.25 mol l$^{-1}$ Mg acetate in MOE solvent. (*j*) Ink I: 0.1 mol l$^{-1}$ acetates in MOE solvent with 2 at% [Fe] in [Mg] + [Fe]. (*k*) Ink J: 0.1 mol l$^{-1}$ acetates in ATA solvent with 10 at% [Sn] in [In] + [Sn]. (*l*) Ink K: thermal chelated ink for ink J, 0.1 mol l$^{-1}$ with 10 at% [Sn] in [In] + [Sn].

## 3.2. Patterns from inkjet printing

Suitable ink is the key to a successful printing process. The printing processes including droplet formation, motion of the droplet, interaction between the liquid droplet and the solid surface of the substrate, drying of the liquid, etc., depend on the properties of the ink, while the target film provides the ultimate judgement for the quality of the ink. The morphologies and features of the printed oxides films are characterized in this part.

As a patterning technique, the reliability of the printing process and the resolution of the printed image are important factors to qualify the printing process. We use Xaar XJ126/50 printhead, representing 126 channels which generate droplets in volume of 50 picolitre, to DOD print different metal oxide films. Ink of SiO$_2$ spheres suspended in FAM (ink A2) is used for printing dots and lines on glass substrates, as the optical microscope images show in figure 2. A dot is printed from a single droplet generated from an individual nozzle of piezoelectric DOD inkjet printhead, while a line is printed from multi-droplets either from the same nozzle or the neighboured nozzles, depending on the target pattern and the motion of the printhead with respect to the substrate. The dots are circular in shape and their diameters are close to each other (90 ± 10 μm), indicating that the volume of the ejected droplets and the interaction between the liquid and solid surface are repeatable, and the printing process is reliable. The number of the aligned droplets resolves the dimension of the pattern, for example, one, two and three drops on the lateral direction for the narrow (85 ± 10 μm), medium (130 ± 10 μm) and wide (260 ± 10 μm) lines shown in figure 2*c*. Considering the nozzle pitch (approx. 137 μm) and the inclination (30°) of the printhead [33], the observed width of the lines can be caused by the wetting of the liquid/solid interface after deposition. Thus, it is important to adjust the

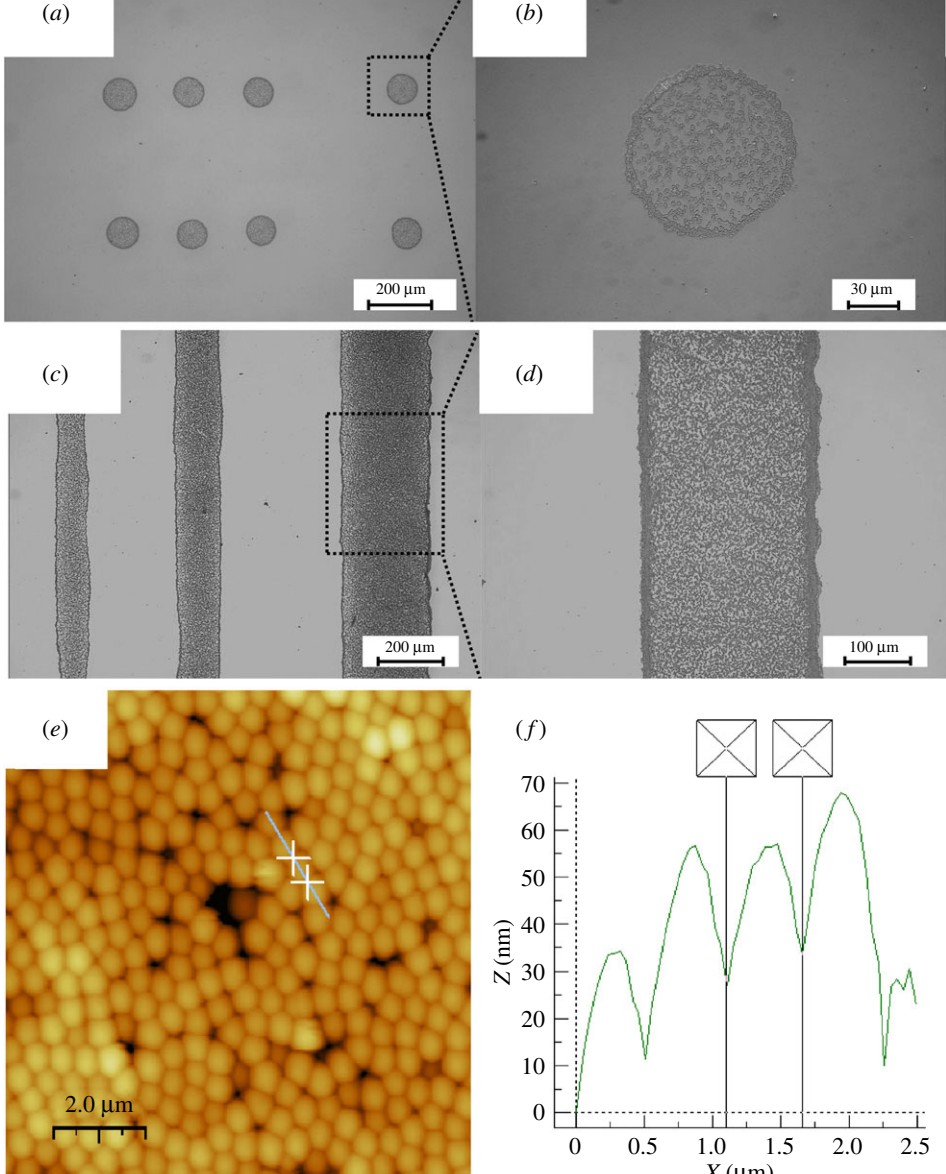

**Figure 2.** Patterns printed from the SiO$_2$ suspensions (ink A2): optical microscopy images of (*a,b*) dots and (*c,d*) lines with different magnifications. (*e*) the AFM image of the printed films and (*f*) a line profile on (*e*).

wettability of the substrate and the surface tension of the ink for a well-controlled patterning process (table 1). For both circles and lines, the SiO$_2$ spheres are observed to accentuate at the contact edge of the patterns, which is known as 'coffee-ring' effect. This effect is well described by Deegan *et al.* [38] from a drop of dried coffee. With a droplet on substrate, a pinned contact line forms at the edge and the liquid flows inside the droplet move the solids from the as-deposited position outwards during the evaporation. By adjusting the wettability of the substrate surface, the viscosity of the liquid, the size and the shape of the solids in the ink, the evaporation rate of the liquid, etc., [27,28,39], the 'coffee-ring' effect can be avoided and uniform films can be prepared for device applications. Figure 2*e* shows the AFM images of the film printed from the particle suspension ink A2. From the line profile, the diameter of a typical sphere is approximately 520 nm, as presented in figure 2*f*.

## 3.3. Films printed from different types of inks

The morphologies and the features are studied for oxide films printed on silicon substrates from different inks. Figure 3 shows a comparison of surface morphology of TiO$_2$ film printed from the suspension ink B, and ZnO

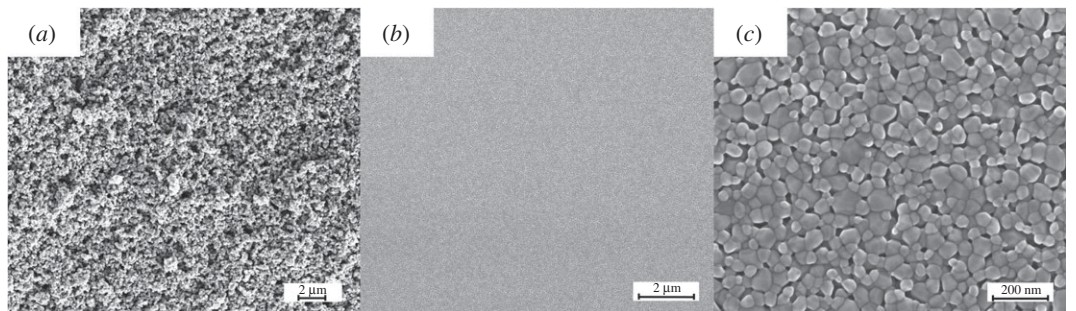

**Figure 3.** SEM surface morphologies of oxide thin films printed from different types of inks: (*a*) TiO₂ film from particle suspensions (ink B) and (*b*,*c*) ZnO film from zinc acetate solution (ink E) with different magnifications.

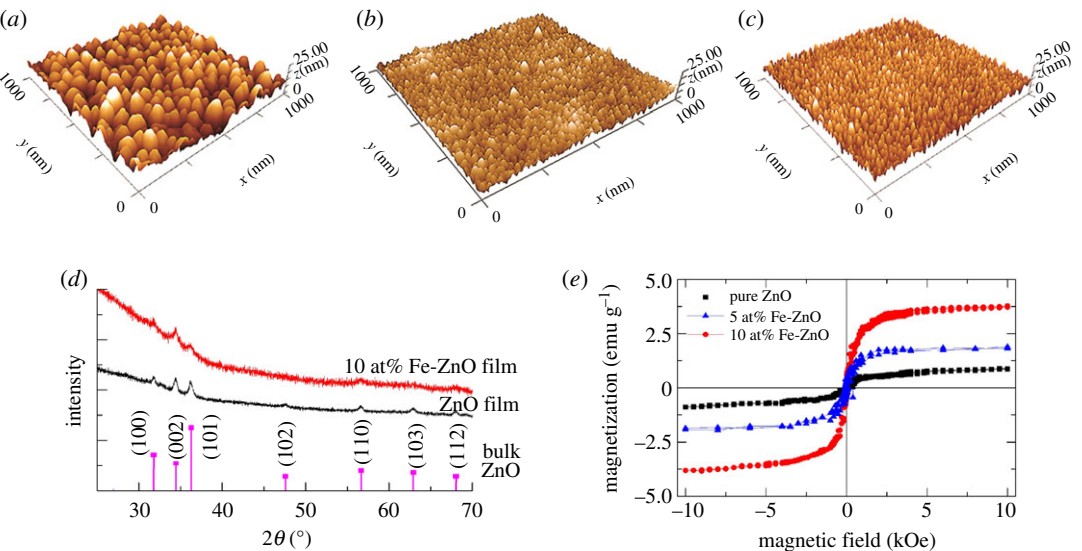

**Figure 4.** *In situ* printed oxide films: three-dimensional AFM images (*a*) ZnO, (*b*) 5 at% and (*c*) 10 at% Fe-doped ZnO thin films. The scanned area is $1 \times 1\ \mu m^2$. (*d*) the XRD patterns of the oxide films comparing with the standard diffraction peaks from the bulk ZnO (JCPDS No. 00-036-1451) [40], and (*e*) the magnetic hysteresis loops of the films. The films for XRD study are prepared on glass substrates, and the other films are on silicon substrates.

film *in situ* prepared from the solution ink E. (i) For the suspension inks, generally high-temperature sintering is required to improve the links among oxide particles after printing. This is an *ex situ* process, from which the oxides are transported to the substrate and form films. Thus, the size and the structure of the particles are difficult to tune. The obtained films are porous with low density and loose with weak bonding among particles. (ii) For acetate solution inks, the as-printed films are acetate precursors, which decompose into the target oxides during the post-calcination process [21]. It is an *in situ* process, where oxide films form *in situ* on substrate. Thus, it is possible to tune the structure (e.g. secondary element doping) and morphology (e.g. grain size) of the target films. The obtained films are uniform and dense with smooth surface, as seen from the high-resolution SEM images presented in figure 3*b*,*c*.

## 3.4. Structure and property tuning of the oxide films

Acetate solution inks can *in situ* produce uniform and dense oxide films, as well as tune the structure and morphology of the films easily. Since multi-acetates can be dissolved in the solution ink, it is possible to dope secondary element into the matrix oxide lattice. Figure 4 shows characterizing of *in situ* printed pure ZnO films and Fe-doped ZnO films from their corresponding acetates solution inks [40]. For doping, the inks are prepared by adding zinc acetate and iron (II) acetate into IPE solvents to obtain solution with two types of cations ($Fe^{2+}$ and $Zn^{2+}$), with total cation concentration of 0.25 mol l⁻¹. The doping concentration of iron can be easily tuned by adjusting the molar ratio of iron acetate to zinc acetate. For instance, 5 at% and 10 at% Fe-doped ZnO films can be prepared from the ink with molar

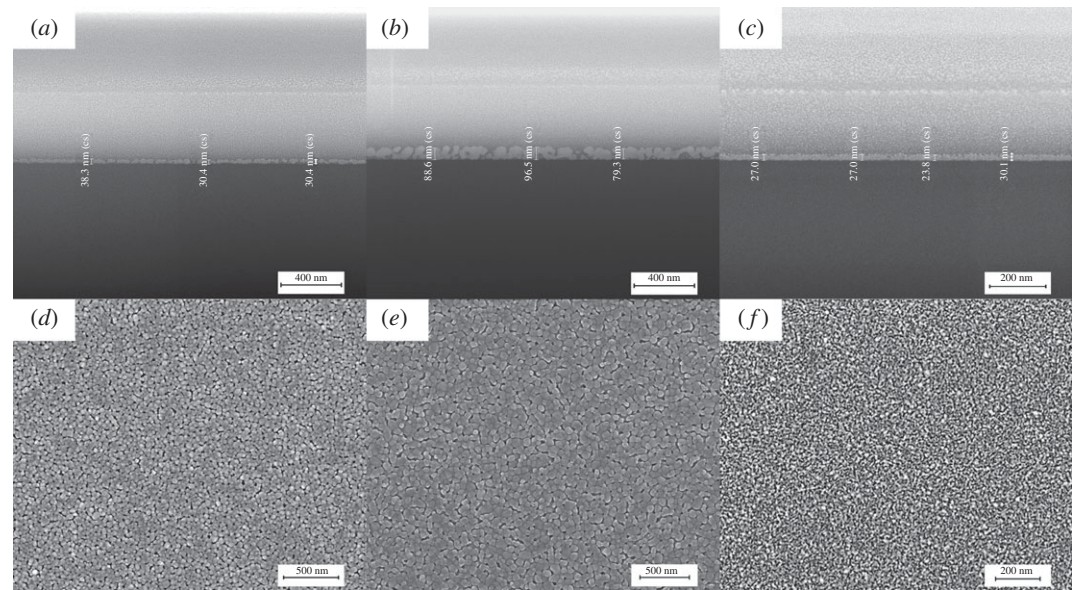

**Figure 5.** SEM images of cross section and surface morphologies of (a,d) single-pass-printed ZnO films, (b,e) three-pass-printed ZnO films and (c,f) single-pass-printed 10 at% Fe-doped ZnO films.

ratio of iron acetate to zinc acetate to be 1 : 19 and 1 : 9, respectively. The annealing temperature can be as low as 400°C, and we use 500°C for 1 hour with optimized properties [40]. All the films are uniform and smooth, with determined root mean squared roughness (Rq) of 3.6, 2.0 and 1.9 nm for pure ZnO, 5 and 10 at% Fe-doped ZnO thin films, respectively (figure 4a–c). With Fe-doping, the crystal grain becomes smaller, which is attributed to the enhanced nucleation of Fe during the *in situ* growth of the oxides. Figure 4d shows the X-ray diffraction peaks of the films which match well with the standard ZnO diffraction peaks. There is no evidence of the presence of iron oxides even with 10 at% Fe-doping, indicating that the Fe ions are incorporated into the matrix ZnO lattice without forming secondary phases [40]. Figure 4e shows the magnetic hysteresis loops of the films. Pure ZnO films show ferromagnetism because of the intrinsic defects, while Fe doping increases the ferromagnetism because of the ferromagnetic ordering of Fe in the films [40–44]. The decrease in grain size would also contribute to the enhancement of ferromagnetism. Acetates solution inks enable an easy and low-cost way for printing component film with tunable structures and properties.

## 3.5. Reliability and repeatability of the printing

The reliability and the repeatability of DOD inkjet printing acetate precursor inks are inspected by the top and cross-section morphologies of oxide films prepared from different inks. For a particular ink, the thickness of the printed film is almost proportional to the amount of printing passes, as the SEM cross-section images show in figure 5a,b for single-pass- and three-pass-printed ZnO films from ink E, respectively. Each printing pass produces approximately 30 nm thick films. The crystals grow up and the films become dense in multi-pass-printed films, leading to a small deviation of the thickness as presented in figure 5d,e. The SEM images of single-pass-printed 10 at% Fe-doped ZnO film from ink F is shown in figure 5c,f. The thickness is approximately 27 nm with reasonable deviation from the undoped ZnO film (figure 5a) because of the smaller particle size and higher density of the doped films. The reliability and the repeatability of inkjet printing oxide films make it possible to design the film with well-controlled structure and properties for functional device applications.

## 3.6. Phase separation

With mixed cations in solution-based ink, phase separation may occur during the post-treatment processes. We observed phase separation for Mg-ZnO and ITO films printed from ink G (figure 1h) and ink J (figure 1k) on glass substrates, as typical images of ITO films (annealed at 500°C for 2 h after printing) shown in figure 6a. The reticular structure is caused by the phase separation of the mixed acetates in the ink since the indium acetate and the tin acetate have different solubility in ATA

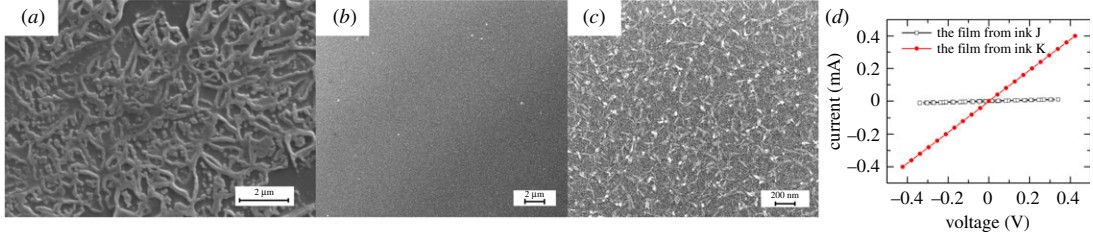

**Figure 6.** ITO thin films: the SEM images of the films printed (*a*) from ink J with micro-reticular structures, (*b*,*c*) from ink K with smooth surface and the nano-feathery structure. (*d*) the I–V curves of the films prepared from ink J and K.

solvent and separate out during the drying process. To obtain uniform ITO films, we chelate the mixed acetates in ATA and prepare ink K. The films prepared from ink K shows feathery nano-structures with smooth surface (figure 6*b*,*c*). Figure 6*d* shows the current–voltage (I–V) curves of the films prepared from different inks. The resistivity ($\rho$) of the thin films can be determined using [7]:

$$\rho = \frac{\pi}{ln2} \frac{V}{I} d,$$

where $d$ is the thickness of the film. They are $1.16\,\Omega\,cm$ and $0.04\,\Omega\,cm$ for the four-pass-printed films (thickness of approx. 80 nm) from ink J and K, respectively. The results indicate that by chelating the mixture of indium acetate and tin acetate with the solvent under reagents and heating, the phase separation of $In_2O_3$ and $SnO_2$ can be effectively suppressed. This could be attributed to the formation of acetylacetonate complexes which act as the precursors of oxides. The conductivity of the films prepared from the chelated inks (ink K) can be dozens of times higher than the films from the as-prepared ink (ink J) because of the nano-feathery structure.

# 4. Conclusion

From the study of dozens of different inks for DOD inkjet printing relevant oxide materials, we observed that design and tailoring of the inks depend on the materials and the applications. The developed acetate solution inks are stable over years with ability of *in situ* printing oxide films directly on substrate. Comparing with the *ex situ* printing process where the oxides are transported to a substrate, the *in situ* process produces dense and smooth films with tunable structure and component, which can be used to print electronic devices. Furthermore, possible phase separation of printing doped films can be suppressed by chelating ligands of cations in the solution-based inks. This work offers guidelines for design of inks for industry inkjet printing of different functional oxide films with controlled thickness and tunable microstructures, and promotes feasibility of inkjet patterning of new materials for device applications.

Ethics. This article does not present research with ethical considerations.

Permission to carry out fieldwork. No fieldwork is required in this work.

Data accessibility. Data available from the Dryad Digital Repository: https://doi.org/10.5061/dryad.n02v6wwsc [45].

Authors' contributions. M.F., K.V.R. and L.B. designed and supervised the experiments. M.F., T.L. and S.Z. carried out the experiments. M.F., T.L., S.Z. and L.B. wrote the manuscript. All authors commented on the manuscript.

Competing interests. We declare we have no competing interests.

Funding. This work was supported by the National Key Research and Development Program of China (2017YFB0305500), National Natural Science Foundation of China (11504055), the Hunan Provincial Natural Science Foundation of China (2018JJ2480) and the Fundamental Research Funds for the Central Universities of Central South University (2019zzts412, 2018zzts324), as well as the Swedish Research Council.

Acknowledgements. We thank Anastasia V. Riazanova, Yan Wu and Voit, Wolfgang for their support of this work.

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
