## [Reviewer comments · Royal Society Open Science]

Review History

RSOS-191652.R0 (Original submission)

Review form: Reviewer 1

Is the manuscript scientifically sound in its present form?

No

Are the interpretations and conclusions justified by the results?

Yes

Is the language acceptable?

Yes

Do you have any ethical concerns with this paper?

No

Have you any concerns about statistical analyses in this paper?

Yes

Recommendation?

Major revision is needed (please make suggestions in comments)

Comments to the Author(s)

The manuscript can be published after major revision.

1# Twelve inks are mentioned in the manuscript. In addition to Figure 1, it is better for authors to provide a table to give a detailed description on the ink, including materials, solvent, surface tension and viscosity.

2# In 2.2, it said "The films are printed on a well cleaned substrate", the name of the substrate should be given. glass? silicon?

3# The diameter of the nozzle should be provided.

4# XRD patterns should be provided. For example, in Figure 3b and c, is the film pure ZnO?

5# Some films, such as ITO, are semiconductor, typically used as transparent conductive films. How about the electrical properties of such film?

6# Some recent papers should be cited in Introduction to emphasize the importance of such technique. For example, <https://doi.org/10.1002/admt.201800546>, J. Mater. Chem. C, 2017,5, 2971-2993

Review form: Reviewer 2

Is the manuscript scientifically sound in its present form?

No

Are the interpretations and conclusions justified by the results?

Yes

Is the language acceptable?

No

Do you have any ethical concerns with this paper?

No

Have you any concerns about statistical analyses in this paper?

No

Recommendation?

Reject

Comments to the Author(s)

This manuscript describes the inkjet printing of various oxide materials and their ink formulations. Although the authors have examined several oxide materials, their characterizations are rather superficial. I suggest more materials property characterizations are necessary, such as line heights, electrical properties, etc. I do not think this manuscript is suitable for the publication at this stage.

Review form: Reviewer 3

Is the manuscript scientifically sound in its present form?

No

Are the interpretations and conclusions justified by the results?

No

Is the language acceptable?

No

Do you have any ethical concerns with this paper?

No

Have you any concerns about statistical analyses in this paper?

No

Recommendation?

Major revision is needed (please make suggestions in comments)

Comments to the Author(s)

The paper contains original material but is written unclearly. I am more inclined towards rejecting this paper but thought of giving them an opportunity to improve the manuscript. I basically agreed with the two Referees that the details of the many oxide materials inks should be carefully described in text. In addition, it is totally unclear about the quality and the performance of the directly-printed oxide thin films. OM (Figure 2) and SEM characterization (Figures 3,5) are obviously not enough for readers. Namely, the defects in inks are not as well characterized, mainly lacking of atomic scale characterizations such as XRD patterns, high-resolution TEM images of microstructure and its diffraction pattern. I strongly suggest the authors make some common microstructural characterizations (including TEM, and Raman spectroscopy). In this respect, a related work on stoichiometric ferrite (Phys. Rev. B 61, 6876 (2000)), should be referenced. Furthermore, there is a lack of understanding about how intrinsic defects (e.g., oxygen vacancies, VO) strongly coupled with properties in general. VO in oxides is ubiquitous and plays an important role in properties of oxides, particularly in form of microstructures. VO-related F-center view and atmospheric annealing-tunable defects mechanism may be possible for diluted magnetism. For example, some recent works (CrystEngComm, 14, 525 (2012); CrystEngComm 15, 2372 (2013); ACS Appl. Mater. Interfaces 6, 4490 (2014); Appl. Phys. Lett. 110, 083107 (2017); Appl. Phys. Lett. 114, 203101 (2019)), which should be referenced in the introduction of the manuscript. Due to its many defects as above, I am sorry to say that I do not recommend publication of this manuscript. The authors should clarify these unclear questions before the paper can be accepted for publication in Royal Society Open Science. I am looking forward to see the revised version.

Decision letter (RSOS-191652.R0)

30-Jan-2020

Dear Dr Fang:

Manuscript ID: RSOS-191652

Title: "Design and Tailoring of Inks for Inkjet Patterning of Metal Oxides"

Thank you for submitting the above manuscript to Royal Society Open Science. Your paper was sent to reviewers and their comments are included at the bottom of this letter.

In view of the concerns raised by the reviewers, the manuscript has been rejected in its current form. However, a new manuscript may be submitted which takes into consideration these comments.

Please note that resubmitting your manuscript does not guarantee eventual acceptance, and that your resubmission will be subject to peer review before a decision is made.

Your resubmitted manuscript should be submitted by 29-Jul-2020. If you are unable to submit by this date please contact the Editorial Office.

On behalf of the Subject Editor Professor Anthony Stace and the Associate Editor Mr Andrew Dunn

REVIEWER(S) REPORTS:

Associate Editor Comments to Author ():

RSC Associate Editor:

Comments to the Author:

I apologise for the delay. Due to conflicting reports, an adjudicator was invited and their report is below (Reviewer 3).

RSC Subject Editor:

Comments to the Author:

(There are no comments.)

Reviewers' Comments to Author:

Reviewer: 1

Comments to the Author(s)

The manuscript can be published after major revision.

1# Twelve inks are mentioned in the manuscript. In addition to Figure 1, it is better for authors to provide a table to give a detailed description on the ink, including materials, solvent, surface tension and viscosity.

2# In 2.2, it said "The films are printed on a well cleaned substrate", the name of the substrate should be given. glass? silicon?

3# The diameter of the nozzle should be provided.

4# XRD patterns should be provided. For example, in Figure 3b and c, is the film pure ZnO?

5# Some films, such as ITO, are semiconductor, typically used as transparent conductive films. How about the electrical properties of such film?

6# Some recent papers should be cited in Introduction to emphasize the importance of such technique. For example, <https://doi.org/10.1002/admt.201800546>, J. Mater. Chem. C, 2017,5, 2971-2993

Reviewer: 2

Comments to the Author(s)

This manuscript describes the inkjet printing of various oxide materials and their ink formulations. Although the authors have examined several oxide materials, their characterizations are rather superficial. I suggest more materials property characterizations are necessary, such as line heights, electrical properties, etc. I do not think this manuscript is suitable for the publication at this stage.

Reviewer: 3

Comments to the Author(s)

The paper contains original material but is written unclearly. I am more inclined towards rejecting this paper but thought of giving them an opportunity to improve the manuscript. I basically agreed with the two Referees that the details of the many oxide materials inks should be carefully described in text. In addition, it is totally unclear about the quality and the performance of the directly-printed oxide thin films. OM (Figure 2) and SEM characterization (Figures 3,5) are obviously not enough for readers. Namely, the defects in inks are not as well characterized, mainly lacking of atomic scale characterizations such as XRD patterns, high-resolution TEM images of microstructure and its diffraction pattern. I strongly suggest the authors make some common microstructural characterizations (including TEM, and Raman spectroscopy). In this respect, a related work on stoichiometric ferrite (Phys. Rev. B 61, 6876 (2000)), should be referenced. Furthermore, there is a lack of understanding about how intrinsic defects (e.g., oxygen vacancies, VO) strongly coupled with properties in general. VO in oxides is ubiquitous and plays an important role in properties of oxides, particularly in form of microstructures. VO-related F-center view and atmospheric annealing-tunable defects mechanism may be possible for diluted magnetism. For example, some recent works (CrystEngComm, 14, 525 (2012); CrystEngComm 15, 2372 (2013); ACS Appl. Mater. Interfaces 6, 4490 (2014); Appl. Phys. Lett. 110, 083107 (2017); Appl. Phys. Lett. 114, 203101 (2019)), which should be referenced in the introduction of the manuscript. Due to its many defects as above, I am sorry to say that I do not recommend publication of this manuscript. The authors should clarify these unclear questions before the paper can be accepted for publication in Royal Society Open Science. I am looking forward to see the revised version.

Author's Response to Decision Letter for (RSOS-191652.R0)

See Appendix A.

RSOS-200242.R0

Review form: Reviewer 3

Is the manuscript scientifically sound in its present form?

Yes

Are the interpretations and conclusions justified by the results?

Yes

Is the language acceptable?

Yes

Do you have any ethical concerns with this paper?

No

Have you any concerns about statistical analyses in this paper?

No

Recommendation?

Accept as is

Comments to the Author(s)

The quality of the revised manuscript has improved; I think it may be now acceptable for publication in Royal Society Open Science.

Decision letter (RSOS-200242.R0)

06-Mar-2020

Dear Dr Fang:

Title: Design and Tailoring of Inks for Inkjet Patterning of Metal Oxides

Manuscript ID: RSOS-200242

It is a pleasure to accept your manuscript in its current form for publication in Royal Society Open Science. The chemistry content of Royal Society Open Science is published in collaboration with the Royal Society of Chemistry.

RSC Associate Editor
Comments to the Author:
(There are no comments.)

Reviewer(s)' Comments to Author:
Reviewer: 3

Comments to the Author(s)
The quality of the revised manuscript has improved; I think it may be now acceptable for publication in Royal Society Open Science.

Appendix A

Response to Reviewer's Comments

We would first like to thank the reviewers for the constructive comments and suggestions, which are very helpful for us to improve the manuscript. We have revised the manuscript by following the suggestions of the reviewers or by addressing the questions raised by the reviewers. The point-to-point responses to all the questions raised are listed as follows. For your convenience, the pages and line numbers of locations of the revision are included in brackets, and changes in the manuscript have been colored in blue.

Reviewer: 1

Comments to the Author(s) The manuscript can be published after major revision.

#1-1 Twelve inks are mentioned in the manuscript. In addition to Figure 1, it is better for authors to provide a table to give a detailed description on the ink, including materials, solvent, surface tension and viscosity.

Response: We have made changes as suggested. Table 1 with detailed information of different inks are provided in the revised paper. (Page 7, Table 1)

#1-2 In 2.2, it said "The films are printed on a well cleaned substrate", the name of the substrate should be given. glass? silicon?

Response: We used both glass and silicon as substrates for printing oxides. Specifically, the SiO₂ and ITO films were printed on glass substrates, and the ZnO-based films were printed on glass substrates for XRD characterizations and on silicon substrates for the other characterizations. We have specified these in the revised paper. (Page 8 line 2, page 11 line 7, page 13 line 2, page 15 the caption of Figure 4, page 16 the last line)

#1-3 The diameter of the nozzle should be provided.

Response: We add this information of the printhead in the revised paper. We used XJ126/50 printhead with 126 channels distributed in 17.2 mm. The nozzle pitch is 137 μm, generated a drop with volume of ~50 picliter. (Page 11 lines 4-6 & 17)

#1-4 XRD patterns should be provided. For example, in Figure 3b and c, is the film pure ZnO?

Response: We have made changes accordingly. XRD diffraction peaks are provided in Figure 4d for both ZnO and 10 at% Fe-doped ZnO films. Figure 3b-c are films of pure ZnO with different magnification. (Page 14 line 17-21, and Page 15 Fig. 4d)

#1-5 Some films, such as ITO, are semiconductor, typically used as transparent conductive films. How about the electrical properties of such film?

Response: We present the I-V curves of the ITO films printed from different inks in the revised paper. (Page 17 lines 6-11 & 15-17, and Fig. 6d)

#1-6 Some recent papers should be cited in Introduction to emphasize the importance of such technique. For example, <https://doi.org/10.1002/admt.201800546>, J. Mater. Chem. C, 2017,5, 2971-2993

Response: We have made changes as suggested. We added these references in the revised introduction part. (Ref. No. 6 and 9).

Reviewer: 2

#2. This manuscript describes the inkjet printing of various oxide materials and their ink formulations. Although the authors have examined several oxide materials, their characterizations are rather superficial. I suggest more materials property characterizations are necessary, such as line heights, electrical properties, etc. I do not think this manuscript is suitable for the publication at this stage.

Response: We thank the referee for his comments on improving our paper. We have made changes as suggested. In the revised paper, we present the property characterizations of the printed oxide films, including AFM image and line profile for SiO₂ film presented in Fig. 1e-f, XRD patterns and magnetic hysteresis loops for (Fe-doped) ZnO films shown in Fig. 4d-e, and I-V curves for ITO films shown in Fig.6d. Related changes have been made in the paper. (Page 12 lines 7-9, Page 14 lines 17-22, Page 15 lines 1-3, Page 17 lines 6-11 & 15-17)

Reviewer: 3

#3-1 The paper contains original material but is written unclearly. I am more inclined towards rejecting this paper but thought of giving them an opportunity to improve the manuscript. I basically agreed with the two Referees that the details of the many oxide materials inks should be carefully described in text.

Response: We thank the referee for giving us the opportunity to improve the manuscript. We have made changes as suggested. The inks for printing oxide films are described carefully in the revised paper with details presented in Table 1. (Page 7, Table 1)

#3-2 In addition, it is totally unclear about the quality and the performance of the directly-printed oxide thin films. OM (Figure 2) and SEM characterization (Figures 3,5) are obviously not enough for readers. Namely, the defects in inks are not as well characterized, mainly lacking of atomic scale characterizations such as XRD patterns, high-resolution TEM images of microstructure and its diffraction pattern. I strongly suggest the authors make some common microstructural characterizations (including TEM, and Raman spectroscopy).

Response: We have made changes accordingly. The properties of the inks are critical for a successful printing process, while the target oxide film provides the ultimately judgment for the quality of the inks. The directly-printed films contain organics and post-annealing is required to get the target oxide films. We focus on the target oxide films, and the micro-structure and property characterizations for final oxide films are provided in the revised paper, including AFM images for SiO₂ film in Fig. 1e-f, XRD patterns and magnetic hysteresis loops for (Fe-doped) ZnO films in Fig. 4d-e, and I-V curves for ITO films in Fig.6d. Instead of using TEM with troubles in sample preparation, we use high-resolution (~10 nm) SEM without sample contamination to characterize the nanometer thick (20~100 nm) oxide films prepared from inkjet printing. For instances, images presented in Fig. 3c, Fig.5 and Fig.6c show nanostructures clearly. In the revised version, the ink properties like the viscosity and surface tension are

provided in Table 1. The defect in the inks could be an interesting topic and we will extend this study in our future work. Related changes have been made in the revised paper. (Page 7 Table 1, Page 12 lines 7-9, Page 14 lines 17-22, Page 15 lines 1-3, Page 17 lines 6-11 & 15-17)

#3-3 In this respect, a related work on stoichiometric ferrite (Phys. Rev. B 61, 6876 (2000)), should be referenced. Furthermore, there is a lack of understanding about how intrinsic defects (e.g., oxygen vacancies, VO) strongly coupled with properties in general. VO in oxides is ubiquitous and plays an important role in properties of oxides, particularly in form of microstructures. VO-related F-center view and atmospheric annealing-tunable defects mechanism may be possible for diluted magnetism. For example, some recent works (CrystEngComm, 14, 525 (2012); CrystEngComm 15, 2372 (2013); ACS Appl. Mater. Interfaces 6, 4490 (2014); Appl. Phys. Lett. 110, 083107 (2017); Appl. Phys. Lett. 114, 203101 (2019)), which should be referenced in the introduction of the manuscript. Due to its many defects as above, I am sorry to say that I do not recommend publication of this manuscript. The authors should clarify these unclear questions before the paper can be accepted for publication in Royal Society Open Science. I am looking forward to see the revised version.

Response: We have made changes as suggested. The effect of intrinsic defect on magnetic properties is discussed, and the references are added in the revised paper. (Page 14 lines 21-22, Page 15 lines 1-3 & Figure 4e, Ref. No. 13, 32, 41-44)